# Growth Performance, Blood Metabolites, Carcass Characteristics and Meat Quality in Finishing Wagyu Crossbred Beef Cattle Receiving Betaine–Biotin–Chromium (BBC) Supplementation

**DOI:** 10.3390/vetsci9070314

**Published:** 2022-06-23

**Authors:** Sukanya Poolthajit, Wuttikorn Srakaew, Theerachai Haitook, Chaiwat Jarassaeng, Chalong Wachirapakorn

**Affiliations:** 1Department of Animal Science, Faculty of Agriculture, Khon Kaen University, Khon Kaen 40002, Thailand; spx_nu177@hotmail.com (S.P.); theerachai.anisci@gmail.com (T.H.); 2Department of Animal Science and Fisheries, Faculty of Agricultural Sciences and Technology, Rajamangala University of Technology Lanna, Nan 55000, Thailand; esso.wutt@gmail.com; 3Division of Theriogenology, Faculty of Veterinary Medicine, Khon Kaen University, Khon Kaen 40002, Thailand; chajar@kku.ac.th

**Keywords:** betaine, biotin, chromium, Wagyu crossbred, marbling

## Abstract

**Simple Summary:**

Increasing the utilization of nutrients in high-producing ruminants such as finishing beef cattle improves growth and meat quality. Feed additives are an alternative way to increasing nutrient utilization, thus improving product quality. Previous studies found that adding feed additives such as betaine–biotin–chromium resulted in increased energy and protein utilization efficiency in growing beef cattle. The purpose of this study was to explore how a combination supplementation affected growth performance, carcass characteristics, and meat quality in finishing Wagyu crossbred beef steers. We found that a combination of betaine–biotin–chromium had a promising response in terms of nutrient utilization resulting in increased fat content in meat. As a result, meat with a relatively high fat content during the finishing period may contribute to improving higher-priced meat and increasing consumer demand.

**Abstract:**

Eighteen Wagyu crossbred steers (average initial body weight: 596.9 ± 46.4 kg; average age: 36 ± 2.7 months) were subjected to three levels of betaine–biotin–chromium (BBC) supplementation for 98 days before slaughter. Animals were fed a basal diet and BBC supplemented at 0, 3 or 6 g/kg of dry matter (DM). The experimental design was a randomized complete block design by a group of animals with six replicates. The intake and digestibility among treatments were not different (*p* > 0.05). The average daily gain (ADG) of steers that received BBC at 6 g/kg of DM (0.79 kg/day) tended to be higher (*p* = 0.07) than that of those receiving BBC at 0 and 3 g/kg/day (0.52 and 0.63 kg/day, respectively). Blood metabolites were not different (*p* > 0.05) among treatments. Carcass characteristic traits included chilled carcass dressing percentage and loin eye area, while meat quality included drip loss, cooking loss, and Warner-Bratzler shear force were not different (*p* > 0.05). Back fat thickness tended to be higher (*p* = 0.07) in steers fed BBC at 6 g/kg. The marbling score did not differ (*p* > 0.05) among treatments; however, the intramuscular fat content of the longissimus dorsi (LD) on a DM basis was significantly higher (*p* < 0.05) in steers fed BBC at 6 g/kg (39.8% DM) than in those fed BBC at 0 g/kg (28.2% DM) and at 3 g/kg (29.1% DM). Based on the findings, BBC supplementation had no effect on growth performance and carcass characteristics; however, BBC at 6 g/kg DM increased fat content in LD muscle of finishing Wagyu crossbred steers.

## 1. Introduction

Marbling, or intramuscular fat deposition in the muscle, is one of the most important characteristics influencing beef quality and palatability [1]. Marbling is associated with beef juiciness, tenderness, and palatability. In the urban market, where pricing is based on the marbling score (MBS), it has a high market price. Demand for such high-quality beef is increasing due to its higher price than local beef. However, the productivity of fattened beef cattle in developing tropical countries has been low due to the animals’ genetic profiles and the provision of feeds and feeding systems based on available low-quality agricultural by-products, such as rice straw [2].

The authors in Park et al. [1] reviewed several factors (genetic, management, and nutritional factors) influencing the marbling score in cattle. MBS is linked to intramuscular fat deposited in muscle [3]. Thailand’s beef producers are still confronted with the issue of variation in beef quality, particularly insufficient marbling levels. Farmers have gradually used Wagyu sires to produce young calves for fattening with the goal of obtaining a high MBS. Wagyu cattle produce more fat in their muscles than other breeds [1,4,5] and have become popular for breeding crossbred fattening cattle rather than European breeds, as they produce a higher MBS. Despite this interest, to the best of our knowledge, only a few studies have been conducted on fattening Wagyu crossbred cattle in Thailand. The authors in Suksombat et al. [6] and Mirattanaphra and Suksombat [7] found that feeding a basal diet (concentrate and rice straw) supplemented with palm and/or linseed oil or protected-rice bran oil to Wagyu crossbred steers had no effect on MBS or fat content in the muscle but improved the fatty acid profiles of the fat deposits in the longissimus dorsi and semimembranosus.

In order for finishing cattle to accumulate intramuscular fat, their net energy consumption must exceed their maintenance and production requirements. As a result, the amount of energy consumed determines the degree of marbling [8,9,10]. One of the most important factors influencing lipogenesis, marbling, and beef quality in ruminants is glucose supply [9]. As a result, gluconeogenesis is critical for the production of glucose, which serves as a universal fuel for cellular, tissue, and whole-animal functions. In the presence of sufficient glucose, insulin increases the activity of acetyl Co-A carboxylase and other enzymes involved in carbon chain elongation [11]. A high-grain diet increases the production of propionic acid in the rumen [12], which increases the supply of glucose to the ruminants because propionic acid serves as a primary precursor in gluconeogenesis [13]. In this regard, methods for increasing gluconeogenesis during the fattening stage of beef cattle may enhance fat deposits in the muscle and adipose tissue.

Feed additives have the potential to enhance intramuscular fat deposits, especially those that stimulate glucose utilization efficiency in muscle. In this regard, the development of products that have potential in animal production and maintain or improve the quality of meat is a challenge. The use of a biotin, betaine, and chromium (BBC) combination in growing bulls appears to be a promising approach to improving nutrient utilization efficiency, resulting in improved bull growth performance [14]. Biotin, chromium, and betaine have been supplemented in ruminant diets as a single source or as a combination of two or more sources, and this was found to have a positive effect on milk yields [15,16]. The authors of Nedjad et al. [17] demonstrated that feeding chromium methionine (Cr-Met) supplementation at 0.4 mg/kg/day to Holstein steers for 4 months during the late fattening period could improve some blood metabolites and beef quality by increasing the polyunsaturated fatty acid (PUFA) and gamma-linoleate composition in beef. The authors of Jin et al. [18] supplemented cattle feed with 1.6 g/kg of rumen-unprotected betaine or 2.9 g/kg of rumen-protected betaine for 68 days and found that the water loss, the redness of the longissimus dorsi muscle, and abdominal fat percentage were reduced compared with the control. However, little is known about biotin in terms of growth performance and meat quality in beef cattle. A recent study by Poolthajit et al. [14] showed that supplementation with a combination of betaine, biotin, and chromium and a diet with a high concentrate and roughage ratio improved nutrient utilization, particularly enhancing the growth rate in growing Thai native bulls. We hypothesized that adding BBC to fattening beef cattle would improve feed utilization, the growth rate, carcass traits, and meat quality. As a result, the purpose of this study was to assess the effect of BBC levels on the growth performance, nutrient utilization efficiency, carcass characteristics, and meat quality of Wagyu crossbred steers, raised in a typical smallholder feeding system during the finishing phases.

## 2. Materials and Methods

### 2.1. Animal Welfare

Animal care was approved by the Animal Care and Use Committee of Khon Kaen University (Record No. IACUC-KKU 83/2562) and procedures were carried out in accordance with the National Research Council of Thailand’s published Ethics of Animal Experimentation.

### 2.2. Experimental Location, Experimental Design, Animals, and Diet

This study was conducted at commercial smallholder farms of the Nong Sung Agricultural Cooperative in Mukdahan Province, Thailand, between September and November 2021. The farms were selected because they are representative of smallholder cattle feedlots in that area. The design of the experiment was a randomized completely block design (RCBD) with smallholder farms set as the blocks. Within each block, six animals were randomized from the same group of animals in the area.

Eighteen Wagyu crossbred (50% Wagyu × 50% (Brahman × Thai Native)) steers in the finishing phase with initial body weight of 596.9 ± 46.4 kg and an average age of 36 ± 2.7 months were used in the feeding trial for 98 days prior to slaughtering. Animals were subjected to Abentel suspension (1 mL/100 kg) (Atlantic Laboratories Corporation, Ltd., Bangkok, Thailand) for intestinal parasites. In order to improve health, vitamins A, D_3_, and E were injected intramuscularly. The animals were adapted to environment, feed, and water supply for 14 days before the 98-day feeding trial. Throughout the experiment, the animals were housed in individual pens (2 m 3 m) with free access to feed and drinking water. Each individual animal in each pen was considered as an experimental unit and was assigned to one of three levels of BBC supplementation. BBC was supplemented at 0, 3, and 6 g/kg DM daily. The BBC contained 100 g/kg of betaine, 0.04 g/kg of biotin, and 0.04 g/kg of chromium picolinate, respectively, so animals received biotin at 0.13 and 0.25 mg/kg DM, Cr at 0.13 and 0.25 mg/kg DM, and betaine at 0.32 and 0.64 mg/kg DM, respectively, for BBC provided at 3 g/kg DM and 6 g/kg DM. Animals were offered the same basal diet, which consisted of concentrate pellet, brewer’s grain, molasses, and rice straw at proportions of 46, 16, 12, and 26% (fresh basis) daily (Table 1). All feed was offered twice daily in two equal portions at 09:00 h and 15:00 h. Refused feed was weighed and recorded daily before the morning feeding.

### 2.3. Data Record, Sample Collection, and Chemical Analysis

Animals were weighed before morning feeding at the beginning and at 38, 68, and 98 days of the experiment to calculate the growth performance. Feed offered and refused was weighed and recorded for each animal on a daily basis. Daily feed intake was calculated by subtracting the amount of feed offered from the amount of feed refused. All feeds and refusals were sampled every month. The DM content of all feed samples was determined by oven drying them at 105 °C to a constant weight, and then the DM was used to calculate the dry matter intake (DMI). Another portion was dried at 60 °C for 72 h and then ground to pass through a 1 mm screen using a Hammer mill for further chemical analysis. Feces samples (approximately 500 g fresh) were collected monthly in the morning by rectal palpation to avoid contamination with soil. Fecal samples were then dried at 60 °C for 72 h and then processed in a similar way to the feed samples for further analysis. The chemical compositions of feed and feces was analyzed for gross energy (GE), ash (method 942.05), crude protein (CP; method 984.13), and ether extracts (EE; method 920.39) using the methods of the Association of Official Analytical Chemists (AOAC) [19]. Fiber fractions such as neutral detergent fiber (NDF) and acid detergent fiber (ADF) were determined according to the method of Van Soest et al. [20] using a fiber analyzer (ANKOM 200, ANKOM Technology, Macedon, NY, USA). Dietary digestibility was measured with acid-insoluble ash (AIA) as an internal marker [21]. All samples were subjected to duplicate analyses. Every month, on the same day as animal weighing, 15 mL of blood was collected from each animal’s jugular vein approximately 2 h after the morning feeding. Blood samples were placed in a sterilized vacuum tube (Greiner Bio-One (Thailand) Ltd., Chonburi, Thailand), packed on ice, and transported to the laboratory for further analysis (Accrediation No. 4138/57; Khon Kaen TLC Lab Center Co. Ltd., Khon Kaen, Thailand). Blood metabolites such as urea-N, glucose, triglycerides, cholesterol, total protein, and albumin concentrations were measured using colorimetric method test kits and an automated analyzer (Roche Diagnostics, Indianapolis, IN, USA) (COBAS INTE-GRA 400 plus analyzer, Roche Diagnostics, USA). Aspartate aminotransferase (AST) and alanine aminotransferase (ALT) were analyzed using kits according to the manufacturer’s protocol (Modular analytic E170, Roche, Germany).

### 2.4. Carcass Characteristics and Meat Quality

At the end of the experimental period, all animals were slaughtered at the standard slaughterhouse at Nong Sung Agricultural Cooperative in Mukdahan Province, Thailand. Carcass characteristics were recorded and evaluated according to Thai Agriculture Commodity and Food Standards for beef. The animals were fasted for 12 h with free water before being transported to the Nong Sung Agricultural Cooperative slaughterhouse. The carcass weights were measured before and after the removal of the head, hide, feet, thoracic organs, internal fats, and abdominal organs. The dressing percentages of warm carcass were calculated as the ratio of warm carcass weight to live weight. Following dressing, the carcasses were transferred to a 4 °C aging room and chilled for 7 days. The chilled carcass weight was recorded and the chilled carcass percentage was calculated similarly to the warm carcass dressing percentage.

Following a 7-day chilling period, muscle samples from the longissimus dorsi (LD) muscle were cut between the 12th and 13th ribs on the right side and transported immediately at 4 °C to the laboratory for physical analysis before being frozen for further chemical analysis.

The loin eye area (LEA) between the 12th and 13th rib surfaces of the LD muscle was measured using a transfer and graph paper while the back fat thickness was measured at a point three-fourths of the LD muscle’s length at the 12th rib. Marbling was evaluated by estimating the amount of intramuscular fat (IMF) visible on the cut surface of the LD muscle between the 12th and 13th ribs using photographic standard scales after chilling for 7 days, with the scoring based on a five-point scale (1 = none, 2 = slight, 3 = small, 4 = moderate, and 5 = abundant) by three well-trained persons according to the Thai Agricultural Commodity and Food Standard [22].

The meat color, L*, a*, and b*, values of three measurement positions of the cut surface of a 3 cm thick LD steak were measured 7 days post-mortem after 30 min of blooming at room temperature using a handheld colorimeter with illuminant D65 and an 8 mm aperture (CR-410 Chroma Meter, Konica Minolta Holdings, Inc., Osaka, Japan). The colorimeter was calibrated by white and black standards before measuring the meat samples. Lightness (L*), redness (a*), and yellowness (b*) were determined following CIE (Commission International de I’Eclairage) color coordinates.

Water-holding capacity was assessed via substance losses occurring during different procedures. Drip loss and cooking losses were determined from 3 cm thick slices of rib meat samples, which were sealed in polyethylene bags. Sample bags were opened and weighed to record the initial weight then stored for 24 h at 4 °C in an automatically controlled refrigerator. Following the storage period, the sample surfaces were dried with soft paper before weighing and recording. Afterwards, every sample that was used for the drip loss measurements was sealed in heat-resistant plastic bags and boiled in a water bath at 80 °C until reaching an internal temperature of 70 °C, then dried with soft paper before weighing and recording for determining the cooking loss. Warner-Bratzler shear (WBS) analysis was conducted according to the American Meat Science Association’s guidelines [23]. Using boiled samples from the cooking loss test, the shear forces were measured after cooling and drying. Meat samples were cut with 1.27 cm diameter cores parallel to the longitudinal orientation of the muscle fibers. Each core was sheared and determined using a WBS force (Stable Micro Systems Ltd., Surrey, UK). Six cores per meat slice were sheared and recorded. A crosshead speed of 200 mm/min and a 5 kN load cell calibrated to read over a range of 0–100 N were applied.

The moisture, crude protein, fat, and ash contents of the LD muscle were determined according to the method described by AOAC [19]. Fatty acid profiles were analyzed according to the method of Folch et al. [24] by incubating them in the form of methyl ester and then quantitatively analyzed by gas chromatography (GC) by chloroform extraction: methanol (2:1). Methylation was achieved with hydrochloric acid and analyzed by GC Agilent Technologies 6890N using an FID probe (flame ionization detector).

### 2.5. Statistical Analysis

All data were subjected to analysis of variance using the generalized linear model procedure of SAS version 9.0 [25]. The data were analyzed using a randomized complete block design, with the model including terms for treatment (df = 2), block (df = 2), and treatment × block (df = 4) according to the following model:Y_ijk_ = μ + ρ_i_ + τ_j_ + ρτ_ij_ + ε_ijk._
where Y_ijk_ is the dependent variable, μ is the overall mean, ρ_i_ is the fixed effect of block (j = 1 to 3), τ_j_ is the fixed effect of dietary treatment (i = 1 to 3), ρτ_ij_ is the interaction between block and dietary treatments, and ε_ijk_ is the residual error. Significance was declared at *p* ≤ 0.05 and tendencies were declared at 0.06 ≤ *p* ≤ 0.10. The values reported are the least squares means and SE of the means generated with the LS Means command and compared using the PDIFF command.

## 3. Results and Discussion

### 3.1. Nutrient Intake and Digestibility

Table 1 shows the chemical composition of the feed used in the experiment, which is the typical feed used for feeding finishing steers in the area. The feed intake and digestibility are shown in Table 2. The DMI and other nutrient intake did not differ (*p* > 0.05) among treatments. The apparent digestion of DM, OM, CP, EE, NDF, and ADF were not influenced (*p* > 0.05) by BBC supplementation. These findings are in accordance with our previous study on Thai Native bulls [14]. We assumed that betaine, biotin, and chromium were utilized in the cellular metabolism rather than used by microorganisms in the rumen; therefore, supplementation of BBC did not show a difference in digestibility among the dietary treatments in this study. However, some studies in the literature have contradicted these findings regarding digestibility. Majee et al. [26] demonstrated that biotin supplementation at 20 mg/day in lactating cows had no effect on DM, OM, or NDF digestibility. Furthermore, Shah et al. [27] found that supplementing dairy cows with betaine (15 g/day) increased the apparent digestibility of DM, OM, CP, NDF, and ADF. Deka et al. [28] discovered that supplementing buffaloes with 1.5 mg Cr/kg DM increased the apparent digestibility of DM, OM, CP, and ADF. Inorganic chromium supplementation did not affect DMI [29] and the digestibility of DM, OM, and CP [30].

### 3.2. Growth Performance and Nutrient Utilization Efficiency

The growth performance and nutrient utilization efficiency of Wagyu crossbred steers supplemented with different levels of BBC are shown in Table 3. After 98 days of feeding, BBC supplementation resulted in a linear increase (*p* < 0.05) in weight gain. The ADG of steers fed 6 g BBC/kg (0.73 kg/day) tended to be higher (*p* = 0.07) than that of steers fed 0 g BBC/kg (0.53 kg/day) and 3 g BBC/kg (0.55 kg). In addition, steers offered 0 g BBC/kg and 3 g BBC/kg had no difference in the feed conversion ratio (FCR) and feed efficiency (FE), but these were significantly lower (*p* < 0.01) in steers fed 6 g BBC/kg. According to the findings of this study, BBC supplementation at 6 g/kg resulted in decreased FCR and increased feed efficiency. Growth performance was higher in steers fed 6 BBC than in steers fed 0 BBC and 3 BBC; this finding concurred with the results observed by Poolthajit et al. [14] in growing Thai Native bulls. It might be postulated that the combination of these additives stimulates glucose synthesis and then uptake into the cells to increase nutrient utilization for gain. There is no conclusive evidence regarding the growth rate of beef cattle fed diets supplemented with one of the single ingredients of BBC, according to the literature. Wang et al. [31] found that supplementing Angus bulls with betaine (0.6 g/kg DM) had no effect on DMI but increased average daily gain and decreased FCR. Furthermore, Kneeskern et al. [32] discovered that chromium supplementation (3 mg/day) had no effect on ADG or FCR in feedlot steers. Biotin (20 mg/day) was given to crossbred cattle (*Bos taurus* × *Bos indicus*), but there was no significant difference in their growth rates [33].

Nutrient utilization efficiency for gain derived from the ratio of nutrient requirements to intake showed that the energy and protein utilization efficiency in steers fed 6 g BBC/kg was higher (*p* < 0.01) than in those fed 0 g BBC/kg and 3 g BBC/kg. This is in agreement with the findings of Poolthajit et al. [14] for Thai Native bulls. Regarding the nutrient requirements for maintenance and weight gain in Wagyu crossbred steers based on the equation reported by Br-Corte Valadares Filho et al. [34] (Table 4), the steers receiving 6 g BBC/kg had energy and protein intakes that were 37.52 and 58.66% above the requirements, respectively, whereas the steers fed with the control diet and 3 g BBC/kg had intakes that were 56.91 and 56.79%, and 75.99 and 74.46% above the requirements, respectively. The efficiency of energy and protein utilization in steers fed BBC at 6 g/kg were 0.68 and 0.63, whereas steers fed BBC at 0 g/kg and 3 g/kg had a similar efficiency of energy and protein utilization (0.59 and 0.57). Because the steers were in the finishing stage, with a very low growth rate, BBC supplementation may not have facilitated protein use in this study [35]. However, excess energy intake over energy requirements for maintenance and gain in steers fed BCC at 6 g/kg (7.35 Mcal ME/day) was lower than that in steers fed BBC at 0 and 3 g/kg (10.33 and 10.41 Mcal ME/day, respectively). Energy intake over that required for maintenance and growth in this study may be stored as fat in subcutaneous and intramuscular adipose tissues [8,9,10].

### 3.3. Blood Metabolites

Table 4 depicts the blood metabolites of Wagyu crossbred steers receiving BBC in different amounts. The plasma urea, triglycerides, cholesterol, total protein, and albumin levels were not significantly different among treatments (*p* > 0.05). Creatinine concentration was higher (*p* < 0.05) in steers supplemented with BBC at 6 g/kg than other treatments, while the globulin in steers fed 6 g BBC/kg was lower (*p* < 0.05) than that in the other treatments. Creatinine is a waste product produced by muscles on a regular basis, whereas globulin is a total protein related to animal health. Although creatinine and globulin concentrations varied between treatments, they were found to be within the normal range in this study [36]. Glucose and insulin concentrations were not different (*p* > 0.05) among dietary treatments. Nejad et al. [17] observed that Cr-Met supplementation did not affect albumin, ALP, calcium, and creatine in the blood of Holstein steers in the late fattening period. In this experiment, glucose and insulin concentrations were not affected by BBC supplementation, which is consistent with our previous study in Thai Native bulls [14]. This concurs with the study of Nejad et al. [17], who found that supplementation with Cr-Met chelate in Holstein steers did not affect serum glucose levels. Previous studies from Kegley et al. [37] also discovered that Cr supplementation had no effect on plasma glucose concentrations in cattle; however, Chang et al. [38] and Stahlhut et al. [39] demonstrated that Cr supplementation reduced plasma glucose concentrations in growing and finishing steers. In dairy cows, Zimmerly and Weiss [40] found that supplementation with biotin had no effect on glucose and insulin concentrations; on the other hand, supplemental biotin linearly increased milk and protein yields. According to Duehlmeier et al. [41], glucose transporter 1 (GLUT1) may be more important than glucose transporter 4 (GLUT4) in ruminant skeletal muscle glucose uptake. This is significant, because GLUT1 is thought to be responsible for basal glucose uptake and GLUT4 is known to be responsible for insulin-stimulated glucose uptake [42], implying that ruminants have higher insulin resistance. It is unclear how diet and management affect insulin sensitivity. It has been demonstrated that dietary Cr supplementation improves insulin sensitivity [43], whereas increasing dietary energy intake had no effect on insulin sensitivity [44]. According to Smith and Crouse [45], glucose is the primary substrate for lipogenesis in IMF tissue, so increased insulin resistance or decreased glucose uptake by the action of GLUT4 in adipose cells would result in less glucose available for fatty acid synthesis. In dairy cows, Zimmerly and Weiss [40] found that supplementation with biotin had no effect on glucose and insulin concentrations; on the other hand, supplemental biotin linearly increased milk and protein yields.

### 3.4. Carcass Characteristics and Meat Quality

Table 5 depicts the carcass characteristics. There were no significant differences (*p* > 0.05) in the warm and chilled carcass weights and percentages, and LEA among dietary treatments. LEA in this study was not affected by BBC supplementation and the average LEA was 95.97 cm^2^. This finding agreed with the findings of Suksombat et al. [6] in Wagyu crossbred steers and of Pimpa et al. [46] in Charolais crossbred steers. However, the LEA in crossbred beef breed in this study was higher than that of dairy crossbred steers (84.49 cm^2^) at a similar slaughter weight reported by Krongpradit et al. [47].

The differences in drip loss, cooking loss, and WBS force among dietary treatments were not statistically significant (*p* > 0.05). Jin et al. [18] found that supplementation with betaine in lambs reduced water loss and shear forces compared with the non-supplemented group. The values of drip loss and cooking loss percentages in this study were inconsistent with values reported earlier in Charolais crossbred steers by Chaiwang et al. [48]. In general, IMF affects the water-holding capacity and chemical composition of meat: as IMF increased from 6.6 to 21.5%, moisture content decreased [49]. As a result, meat with a high IMF had lower drip loss and water loss during cooking [50]. These findings confirm that IMF can influence meat tenderness, despite the fact that the contribution of IMF to meat tenderness has been widely debated [51]. As a result, increased IMF content could improve meat’s water-holding capacity [52]. The WBS forces in the LD muscle were unaffected (*p* > 0.05) by the addition of BBC to the diet (Table 5). These WBS forces were higher than those reported by Suksombat et al. [6] in Wagyu crossbred steers (3.57–3.63 kg/cm^2^), but lower than those reported by Pimpa et al. [46] in Charolais crossbred steers (5.94–9.01 kg/cm^2^). The average WBS force was close to 5.37 kg/cm^2^, which exceeded the value allowing the meat to be classified as tough [53].

The MBS was not significantly different (*p* > 0.05) among treatments, while back fat thickness in steers supplemented with BBC at 6 g/kg DM tended to be higher (*p* = 0.07) than in steers supplemented with other BBC levels. However, we observed that the MBS has a close relationship with fat content in muscle (*r* = 0.74, *p* < 0.01). The authors of Kruk et al. [3] demonstrated that factors influencing assessments of marbling were %IMF and breed, but other traits, such as eye muscle area, melting point, fat color, and meat color, were not significant.

Meat color in this study, in terms of lightness (L*), redness (a*), and yellowness (b*), was not different (*p* > 0.05) among treatments (Table 5). The authors of Wulf and Wise [54] found that lean maturity was highly correlated with the values of the color parameters L*, a*, and b*. Furthermore, the b* value has been found to be highly correlated with the thickness of back fat [55]. Similarly, high b* values resulted in high back fat thickness (1.66–2.33 cm) in the current study, compared with the b* and back fat thickness in Wagyu crossbred steers reported by Suksombat et al. [6] and Mirattanaphra and Suksombat [7]. The b* values and back fat thickness were higher than in previous studies in dairy steers [47] and Charolais crossbred steers [46]. This is probably due to a higher excess energy intake that was used for fat synthesis.

Table 6 shows the meat composition of the LD muscle. Moisture, fat, and ash content did not differ (*p* > 0.05). The CP content decreased linearly (*p* < 0.01) and was lowest (*p* < 0.05) in meat from steers given 6 g BBC/kg DM. Because moisture contents vary in meat samples, calculating nutrient content on a DM basis may be a good method to use because it yields specific information on a more uniform basis. On a DM basis, the CP and ash contents were not different (*p* > 0.05), but the fat content was higher (*p* < 0.05) in steers given 6 g BBC/kg DM compared with that in those fed 3 g BBC/kg DM and 0 g BBC/kg DM. Pimpa et al. [46] found that fat content on a DM basis in the meat of fat-supplemented cattle was higher than in that of non-supplemented cattle. In this study, an increase in fat content in the meat resulted in a decline in CP and moisture contents in meat, with a negative correlation to CP (*r* = 0.88, *p* < 0.01) and moisture (*r* = 0.80, *p* < 0.01). This finding was supported by Pimpa et al. [46] who reported that when intramuscular fat increased from 7.9 to 11.8%DM, moisture content decreased.

The uptake of excess net energy, according to Park et al. [1], is a critical component in IMF deposition. When dietary energy increases, the fat accumulation mainly increases due to lipogenic gene expression and decreased lipolytic gene expression in the adipose tissue [10]. In this study, steers under all dietary treatments had an energy intake above their requirements for maintenance and gain (average = 1.50), with the fat content in the muscle averaging 33.14%. This is in line with the findings of Krongpradit et al. [47], who discovered that dairy steers received more excess energy intake (average 1.4) and had more fat in the muscle (average 28.47%). Similarly, Pimpa et al. [46] found that including 5% fat in the diet increased energy intake above the requirements, resulting in a 41.98% increase in muscle fat percentage. The authors of Suksombat et al. [6] and Mirattanaphra and Suksombat [7] found that adding oil to the diets fed to Wagyu crossbred steers had no effect on muscle fat content, which was most likely due to energy intake being lower than animal requirements.

Increased BBC supplementation, on the other hand, resulted in increased fat content in the LD muscle, despite having no effect on the MBS. Aside from the breed factor, this finding may be explained by BBC supplementation, specifically biotin and chromium, stimulating adipocytes in the muscle to take up more glucose for de-novo fat synthesis in steers [56]. The authors of Wu et al. [57] recently demonstrated that betaine regulates lipid metabolism in adipogenic differentiated skeletal muscle cells via the ERK1/2-PPAR (gamma) signaling pathway, thereby increasing fat deposition. However, the mechanism by which betaine regulates the lipid metabolism in skeletal muscle cells is unknown. Moreover, fat content in the muscle could be altered by nutritional factors [1]. Indeed, Pimpa et al. [46] added 5% fat to the diet of steers and found that fat content in the muscle was increased by 34.4 to 49.6%. However, the results of Suksombat et al. [6] and Mirattanaphra and Suksombat [7] showed no effect on the fat content in the muscle when palm or rice bran oil was added at 100 g to 200 g/day but produced changed fatty acid profiles.

The intramuscular fatty acid content was not significantly different (*p* > 0.05) among BBC supplementation treatments (Table 7). Fatty acids in the muscle were dominated by monounsaturated fatty acids (MUFA) at 50.55% (49.97–50.94%), followed by saturated fatty acids (SFA) at approximately 47.65% (47.27–48.27%) and polyunsaturated fatty acids (PUFA) at approximately 1.77% (1.71–1.82%). The mean n-6:n-3 ratio was 6.54, which was higher than the 4.0 value recommended by nutritional advice [6]. Fattening cattle fed a high-concentrate diet may have a higher proportion of PUFA, dominated by n-6, particularly C18:2n6c. The authors of Suksombat et al. [6] added 200 g of linseed oil, which resulted in a decreased n-6:n-3 ratio from 9.97 to 2.07 in the longissimus dorsi muscle. The mean values of the unsaturated fatty acids (UFA):SFA, MUFA:SFA, and PUFA:SFA ratios in the current study were 1.12, 1.06, and 0.04, respectively, which are normal values in beef, according to Nejad et al. [17].

Oleic acid (C18:1n9c) was found to be the most abundant fatty acid in the IMF in this study. The high proportion of oleic acid could be attributed to feeding the steers brewer’s grain and high-concentrate ratio diets. Brewer grain contains a high percentage of linoleic acid (C18:2) (50.17%) [58], which would be converted to oleic acid in the rumen via biohydrogenation and accumulate in muscles [59]. In general, the high proportion of oleic acid could be attributed to a high-starch diet [48]. Steers fed grain-based finishing diets typically have lower SFA and higher MUFA levels [60]. Beef loins with high IMF had high tenderness, juiciness, and flavor scores [61]. Because oleic acid is thought to be an umami component in beef, Hanwoo cattle have consistently been reported to produce highly palatable, high marbling beef, with a high oleic acid content [62]. The authors of Hwang et al. [63] clearly demonstrated that oleic acid is a more influential factor in the taste of Hanwoo beef than fat content. In this regard, Jung et al. [64] proposed that increasing the IMF content in beef can improve overall palatability by increasing the tenderness, flavor, and/or juiciness.

## 4. Conclusions

Based on these study results, BBC supplementation had no effect on intake, digestibility, growth performance, or carcass characteristics of finishing Wagyu crossbred cattle. However, BBC supplementation at 6 g/kg DM increased fat content (on a DM basis) in LD muscle compared with control treatments. More research into the influence of BBC on the meat quality of beef cattle should be conducted.

## Figures and Tables

**Table 1 vetsci-09-00314-t001:** The nutrient contents of concentrate pellet, brewer grain, molasses, rice straw, and BBC used in experiment.

Item	Concentrate Pellet	Brewer’s Grain	Molasses	Rice Straw	BBC
Dry matter (DM), %	96.6	23.9	72.3	93.2	98.1
Organic matter (OM), % DM	91.7	96.3	85.7	88.9	49.3
Crude protein (CP), % DM	14.6	34.7	5.8	4.9	5.3
Ether extracts (EE), % DM	3.0	6.0	1.4	1.1	-
Neutral detergent fiber (NDF), % DM	25.4	54.8	0.3	82.5	51.2
Acid detergent fiber (ADF), % DM	13.1	17.6	0.1	48.8	33.3
Ash, % DM	8.3	3.2	14.3	11.1	50.7
Gross energy (GE), Mcal/kg DM	4.03	5.29	3.57	3.94	2.30
Metabolizable energy (ME), Mcal/kg DM	2.80	2.87	2.83	1.58	1.60

BBC = betaine–biotin–chromium.

**Table 2 vetsci-09-00314-t002:** Feed and nutrient intake and apparent digestibility in steers fed varying levels of BBC supplementation.

Item	BBC Levels, g/kg			SEM	Contrast ^1^, *p*-Value	
	0 BBC	3 BBC	6 BBC		L	Q
Feed intake, kg DM/d						
Concentrate	5.92	6.00	5.64	0.15	0.22	0.25
Brewer’s grain	1.45	1.40	1.47	0.04	0.73	0.24
Molasses	1.41 ^a^	1.45 ^a^	1.21 ^b^	0.02	<0.01	<0.01
BBC	0.00 ^a^	0.04 ^b^	0.07 ^c^	0.001	<0.01	0.63
Rice straw	2.90	2.86	2.75	0.10	0.33	0.76
Total	11.68	11.75	11.13	0.22	0.11	0.22
% of BW	1.81	1.86	1.84	0.04	0.61	0.61
g/kg BW ^0.75^	91.33	93.14	91.35	1.83	0.99	0.44
C:R ratio	0.75	0.76	0.75	0.64	0.82	0.59
Nutrient intake, kg/d						
Organic matter	10.62	10.67	10.11	0.19	0.09	0.23
Crude protein	1.59	1.59	1.54	0.03	0.15	0.41
Ether extract	0.53	0.53	0.51	0.01	0.29	0.39
Neutral detergent fiber	5.12	5.09	4.99	0.12	0.45	0.81
Acid detergent fiber	2.53	2.53	2.43	0.06	0.28	0.55
GEI, Mcal/d	48.01	48.17	45.68	0.88	0.09	0.25
MEI, Mcal/d	29.25	29.55	27.68	0.69	0.14	0.23
Apparent digestibility, %						
Dry matter	72.35	72.49	70.78	1.29	0.41	0.57
Organic matter	76.47	76.57	75.04	1.36	0.47	0.63
Crude protein	72.58	73.57	70.53	1.67	0.41	0.35
Ether extract	92.93	92.86	92.39	0.56	0.51	0.78
Neutral detergent fiber	50.31	50.68	50.92	2.09	0.84	0.98
Acid detergent fiber	25.53	27.27	25.10	2.82	0.92	0.59
Gross energy	74.22	74.91	73.81	1.58	0.86	0.65

BBC = betaine–biotin–chromium combination, 0 BBC = control with no BBC supplementation, 3 BBC = BBC supplementation at 3 g/kg DM, and 6 BBC = BBC supplementation at 6 g/kg DM, BW = body weight, C:R ratio = concentrate to roughage ratio, GEI = gross energy intake, MEI = metabolizable energy intake, L = linear effect, Q = quadratic effect, SEM = standard error of the mean. ^1^ Orthogonal polynomial contrast. ^a,b,c^ Means in the same row with different superscripts were differed significantly (*p* < 0.05).

**Table 3 vetsci-09-00314-t003:** Growth performance and nutrient efficiency in steers fed varying levels of BBC supplementation.

Item	BBC Levels, g/kg			SEM	Contrast ^1^, *p*-Value	
	0 BBC	3 BBC	6 BBC		L	Q
Initial BW, kg	616.17	605.50	569.00	15.41	0.06	0.51
Final BW, kg	668.17	660.50	640.67	14.82	0.22	0.75
BW gain, kg	52.00	55.00	71.67	5.56	0.03	0.34
ADG, kg/day	0.53	0.56	0.73	0.06	0.03	0.35
FCR, kg feed/kg gain	20.66	18.08	12.94	2.26	0.04	0.66
FE, kg gain/kg feed	0.06 ^a^	0.06 ^a^	0.09 ^b^	0.006	<0.01	0.22
Energy efficiency						
Requirement						
ME for maintenance	12.49	12.35	11.93	0.22	0.10	0.61
ME for gain	5.64	5.94	7.62	0.56	0.03	0.35
Total	18.13	18.29	19.55	0.58	0.12	0.46
ME intake	29.25	29.55	27.68	0.69	0.14	0.23
Efficiency, Req/Intake	0.59 ^a^	0.59 ^a^	0.68 ^b^	0.02	<0.01	0.08
Protein efficiency						
Requirement						
CP for maintenance	0.66	0.66	0.63	0.01	0.09	0.55
CP for gain	0.24	0.26	0.34	0.03	0.03	0.35
Total	0.90	0.91	0.97	0.04	0.14	0.54
CP intake	1.59	1.59	1.54	0.03	0.15	0.41
Efficiency, Req/Intake	0.57 ^a^	0.57 ^a^	0.63 ^b^	0.02	0.02	0.11

BBC = betaine–biotin–chromium combination, 0 BBC = control with no BBC supplementation, 3 BBC = BBC supplementation at 3 g/kg DM, and 6 BBC = BBC supplementation at 6 g/kg DM, BW = body weight, ADG = average daily gain, FCR = feed conversion ratio, FE = feed efficiency, ME = metabolizable energy, CP = crude protein, L = linear effect, Q = quadratic effect, SEM = standard error of the mean. ^1^ orthogonal polynomial contrast. ^a,b^ Means in the same row with different superscripts were differed significantly (*p* < 0.05).

**Table 4 vetsci-09-00314-t004:** Blood metabolites in steers fed varying levels of BBC supplementation.

Item	BBC Levels, g/kg			SEM	Contrast ^1^, *p*-Value	
	0 BBC	3 BBC	6 BBC		L	Q
BUN, mg/dL	15.92	14.75	15.83	1.51	0.97	0.56
Total protein, g/dL	7.11	7.38	6.93	0.14	0.40	0.06
Creatinine, mg/dL	1.96 ^a^	1.74 ^ab^	2.12 ^b^	0.08	0.21	0.02
Albumin, g/dL	4.08	3.84	4.13	0.10	0.68	0.05
Globulin, g/dL	3.03 ^a^	3.53 ^ab^	2.83 ^b^	0.17	0.41	0.02
Cholesterol, mg/dL	202.58	177.42	212.83	19.20	0.71	0.23
Triglycerides, mg/dL	37.83	32.75	37.58	3.75	0.96	0.31
AST, U/L	38.92	41.33	39.33	3.21	0.93	0.59
ALT, U/L	16.17	15.33	16.58	1.04	0.78	0.43
Glucose, mg/dL	70.42	68.83	69.08	1.03	0.39	0.49
Insulin, uIU/mL	10.56	10.92	6.63	1.92	0.18	0.35
Glucose:Insulin ratio	0.07	0.06	0.10	0.04	0.33	0.67

BBC = betaine–biotin–chromium combination, BUN = blood urea nitrogen, AST = aspartate aminotransferase, ALT = alanine aminotransferase. 0 BBC = control with no BBC supplementation, 3 BBC = BBC supplementation at 3 g/kg DM, and 6 BBC = BBC supplementation at 6 g/kg DM, L = linear effect, Q = quadratic effect, SEM = standard error of the mean. ^1^ orthogonal polynomial contrast. ^a,b^ Means in the same row with different superscripts were differed significantly (*p* < 0.05).

**Table 5 vetsci-09-00314-t005:** Carcass characteristics in steers fed varying levels of BBC supplementation.

Item	BBC Levels, g/kg			SEM	Contrast ^1^, *p*-Value	
	0 BBC	3 BBC	6 BBC		L	Q
Slaughter weight, kg	663.17	655.50	635.67	14.82	0.22	0.75
WC weight, kg	379.50	370.33	367.33	9.16	0.37	0.79
WC dressing percentage, %	57.26	56.55	57.74	0.89	0.70	0.41
CC weight, kg	368.49	359.96	356.31	9.12	0.36	0.78
CC dressing percentage, %	55.57	54.91	56.05	0.88	0.74	0.41
Back fat thickness, cm	1.78	1.60	2.33	0.20	0.08	0.09
Loin eye are, cm^2^	97.12	91.55	99.24	3.96	0.71	0.20
Marbling score	2.67	2.50	3.00	0.22	0.30	0.24
Meat color						
L* (lightness)	36.54	36.81	36.91	1.31	0.84	0.96
a* (redness)	23.79	25.06	24.42	0.90	0.46	0.73
b* (yellowness)	14.06	14.48	14.41	0.39	0.54	0.62
Drip loss, %	2.67	2.56	2.68	0.14	0.97	0.51
Cooking loss, %	21.99	19.82	19.91	1.16	0.24	0.49
WBS, kg/cm^2^	5.24	5.95	5.82	0.93	0.67	0.72

BBC = betaine–biotin–chromium combination, WC = warm carcass, CC = chilled carcass, WBS = Warner-bratzler shear force, 0 BBC = control with no BBC supplementation, 3 BBC = BBC supplementation at 3 g/kg DM, and 6 BBC = BBC supplementation at 6 g/kg DM, L = linear effect, Q = quadratic effect, SEM = standard error of the mean. ^1^ orthogonal polynomial contrast.

**Table 6 vetsci-09-00314-t006:** Meat composition in steers fed varying levels of BBC supplementation.

Item	BBC Levels, g/kg			SEM	Contrast ^1^, *p*-Value	
	0 BBC	3 BBC	6 BBC		L	Q
Moisture, %	65.88	67.35	66.07	1.66	0.94	0.51
CP, %	22.72 ^a^	21.51 ^ab^	20.33 ^b^	0.49	<0.01	0.98
Fat, %	9.88	9.90	13.56	1.50	0.12	0.35
Ash, %	1.03	1.03	0.93	0.04	0.18	0.41
On a DM basis						
CP, % DM	67.80	67.01	59.99	3.06	0.10	0.43
Fat, % DM	28.17 ^a^	29.13 ^a^	39.81 ^b^	3.10	0.03	0.23
Ash, % DM	3.07	3.22	2.75	0.22	0.35	0.28

BBC = betaine–biotin–chromium combination, CP = crude protein, 0 BBC = control with no BBC supplementation, 3 BBC = BBC supplementation at 3 g/kg DM, and 6 BBC = BBC supplementation at 6 g/kg DM, L = linear effect, Q = quadratic effect, SEM = standard error of the mean. ^1^ orthogonal polynomial contrast. ^a,b^ Means in the same row with different superscripts were differed significantly (*p* < 0.05).

**Table 7 vetsci-09-00314-t007:** Fatty acid profiles in LD meat in steers fed varying levels of BBC supplementation.

Item	BBC Levels, g/kg			SEM	Contrast ^1^, *p*-Value	
	0 BBC	3 BBC	6 BBC		L	Q
Fatty acids, % Fat						
Lauric acid (C12:0)	0.12	0.13	0.11	0.01	0.50	0.19
Myristic acid (C14:0)	3.91	3.54	3.98	0.28	0.87	0.27
Myristoleic acid (C14:1)	1.17	1.19	1.44	0.09	0.06	0.33
Pentadecanoic acid (C15:0)	0.29	0.27	0.31	0.01	0.25	0.06
Palmitic acid (C16:0)	31.09	31.05	31.31	0.66	0.81	0.85
Palmitoleic acid (C16:1)	4.73	4.34	4.86	0.17	0.60	0.06
Stearic acid (C18:0)	11.79	12.06	12.32	0.32	0.28	0.99
Oleic acid (C18:1n9c)	44.85	45.41	43.66	0.96	0.41	0.36
Linoleic acid (C18:2n6c)	1.26	1.27	1.11	0.08	0.21	0.36
Linolenic acid (C18:3n3)	0.24	0.23	0.25	0.02	0.68	0.66
Arachidic acid (C20:0)	0.09	0.10	0.12	0.03	0.57	0.50
cis-11-Eicosenoic acid (C20:1)	0.11	0.11	0.12	0.01	0.62	0.48
cis-8,11,14-Eicosatrienoic acid (C20:3n6)	0.11	0.12	0.12	0.02	0.31	0.76
Arachidonic acid (C20:4n6)	0.18	0.19	0.24	0.04	0.39	0.35
Behenic acid (C22:0)	0.04	0.04	0.03	0.01	0.22	0.80
Lignoceric acid (C24:0)	0.05	0.05	0.05	0.01	0.91	0.46
SFA	47.42	47.27	48.27	0.85	0.51	0.58
UFA	52.65	52.87	51.80	0.85	0.51	0.58
MUFA	50.75	50.94	49.97	0.78	0.49	0.56
PUFA	1.79	1.82	1.71	0.13	0.76	0.85
Σn-3	0.24	0.23	0.25	0.02	0.68	0.66
Σn-6	1.55	1.59	1.47	0.11	0.70	0.80
Σn-6:Σn-3	6.58	6.86	6.19	0.41	0.62	0.58
UFA:SFA	1.12	1.12	1.08	0.04	0.48	0.58
MUFA:SFA	1.08	1.08	1.04	0.04	0.55	0.62
PUFA:SFA	0.04	0.04	0.04	0.003	1.00	1.00

BBC = betaine–biotin–chromium combination, SFA = saturated fatty acids, UFA = unsaturated fatty acids, MUFA = monounsaturated fatty acids, PUFA = polyunsaturated fatty acids. 0 BBC = control with no BBC supplementation, 3 BBC = BBC supplementation at 3 g/kg DM, and 6 BBC = BBC supplementation at 6 g/kg DM, L = linear effect, Q = quadratic effect, SEM = standard error of the mean. ^1^ orthogonal polynomial contrast.

## Data Availability

Not applicable.

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
