# Peer review of "Growth Performance, Blood Metabolites, Carcass Characteristics and Meat Quality in Finishing Wagyu Crossbred Beef Cattle Receiving Betaine–Biotin–Chromium (BBC) Supplementation"

_vetsci, 2022, doi:10.3390/vetsci9070314_

Round 1

Reviewer 1 Report

Authors described how betain biotin chromium could affect to growth and productive parameters of Wagyu crosbreeds steers. I think that the sample size is extremely low (n=18), and then they have three levels of BBC, thus, in each group there are just 6 steers. Moreover, there are some carcass measurements that I found unclear. 

Apart from this two big issues, I have some other comments.

General comments:

  • Abstract is too long, try to shorten it.
  • ln 15. BBC is the first time you use it, please say complete altough you said it in the tittle.
  • Avoid starting sentences with acronyms, please follow this comment throughout the manuscript.
  • ln 111-112. Specify which anti-parasites. 
  • ln 113. Explain that "adaptation time"
  • figure 1. I guess that the units are all percentages, except GE and Me, but please include them
  • ln 150. really jugular? did not you use the coccigeal vein?
  • ln 153. how much time did you spend from obtaining the sample and getting the plasma? Ideally, it must be centrifugated when obtained, and then store at -20, or even -80, if process is not at the moment.
  • ln 155. colorimetric: did you mean that the measurements were performed with a spectophotometer? pleasy specify
  • ln 159. Did not you measure leptin? bhb? fructose? lactate? did you difference between HDL and LDL?
  • ln 172-179. why did not you use ultrasound scanner as it is described? do you have monthly data of the ultrasound results in these steers? Moreover, find references to support the measurements you took, apart from the 22.
  • ln 226. did you apply a normality test? which are the results?
  • ln 241. I am not sure if results and discussion could be shown together, please check journal guidelines
  • ln 253 - 255. in which breeds? Wagyu?
  • ln 258. the same question than for the previous comment.
  • table 2. I guess it is obvious that differences for BBC are in the fed... I would delete them. What is BG? How could you explain molasses differences?
  • ln 282-286. please specify breeds and even ages, because it is not the same showing results form other studies with holstein steers of 13 months of age or Black Japanese with 38 months of age.
  • I would place table 3 on line 288, then it paragraph and table 4.
  • ln 322. It is table 3. You are going back and forward, please review this point
  • ln 322-352. You are exposing some studies related to glucose and insulin. If you find no differences through BBC levels, why did you focus on it? There were differences due to creatinine levels and globulins, why do not you try to explain why?
  • line 353. In this part, avoid repeating results. We can see all that you describe in the table, which I think it is perfect. Do not repeat it all in the paragraphs, just try to explain why the back fat thickness is different trough groups.
  • ln 406. Please, apply what said just above...
  • where is table 7 mentioned in the manuscript?
  • I would shorten all of the fatty acids profile. Too much tables and information, try to mix them and reduce them, specially if there are no statistical differences.
  • table 8. When you say p value = 0.07, is it for the IMF results? or the BF results? When you put ** it means statistical differences and then you place superscripts, I did not find them...
  • Could not you mix table 8 and 9? The same of the superscripts happen in table 9
  • I would like to re-analyse conclusions when having all the results data clear. 

Reviewer 2 Report

The study describes the impacts on carcass/meat quality and composition resulting from supplementation to Wagyu crossbred beef cattle in Thailand.  Several details are lacking from the methodology to make this repeatable and thus not yet suitable for publication.  Specific comments are detailed below.

Ln15:  BBC should be defined in the abstract before using the abbreviation.

Ln23: if you’re referring to dressing percentage for Hot and Chilled Carcass percentage, then specify dressing percentage.  If you are referring to weight, them specify weight.  It is not clear what is meant by hot carcass percentage.

Ln24: Cooking loss and Warner-Bratzler shear force are not carcass characteristic traits, as they are not assessed on the carcass, but rather on meat obtained from the carcass.

Ln25: it is not appropriate to say that a trait “slightly increased” when it isn’t significant.  In fact, it wasn’t even a tendency. 

Ln26-28: IMF had to be manipulated to a DM basis to see significant differences.  This must be specified, because on a raw percentage basis there were not differences in IMF. 

Ln 31-32: this statement is not accurate.  There was only a tendency to improve growth and there were no differences in marbling, IMF, or shear force – the 3 main factors investigated on meat quality.  This statement must be revised to reflect the actual outcomes of the study. 

Ln36-37: this statement must have references

Ln109: please describe Thai Native cattle.  Was a recognized breed utilized?  Please specify.

Ln177-178:  Are the photographic standard scales based on another countries beef quality grading system, perhaps USDA since the nomenclature aligns with the marbling categories, Meat Standards Australia, or Japan.  When searching for the Standards, internet searches do not produce photographic standards (or even the standards themselves), so it would be helpful to have reference for marbling scores. 

Were marbling scores called in whole numbers (1 or 2 or 3 or 4 or 5) or were incremental marbling scores used (i.e. 1.3 or 4.5 or 3.2)?  This should be specified.  Was marbling score assessed by a trained professional (i.e. beef carcass grader), someone from the research team, or plant personnel?

Ln181 & 186: dip should be drip

Ln181-182: where did these slices of rib meat come from?  You never describe meat collection.  Was the entire ribeye roll (cube roll) removed and them sliced into 2.54-cm steaks?  Was a smaller portion removed and from what area?  Was the meat obtained on day 7 postmortem immediately after grading or at a later time?  Specify which muscle(s) were in the “slices of rib meat”.

Ln182 & 186: how was meat sealed?  In a vacuum sealed bag or Ziploc type bag??

Ln189: was WBSF conducted on fresh steaks (not frozen and thawed)?  Were these steaks collected at 7 days postmortem or some other time?  How was meat stored if samples were not tested immediately after collection?

Ln197: longissimus dorsi is too vague – was it longissimus thoracis or lumborum?  Specifically what part of the loin was meat derived?  This can have a significant impact on composition.

Ln239-240: These results are stuck in random locations within the results.  If you are going to include the analysis, the coefficients should be compiled into a table and discussed when appropriate.

Ln381-398: these 2 paragraphs should be moved up to ln 362.  These are traits assessed on the carcass.  They are out place. 

Ln381: this statement is false.  It wasn’t even a tendency so it is inappropriate to say it was “slightly increased”.

Ln389-398:  I’m not sure of the relevance about talking about correlations of instrumental color  with backfat in other papers when there were not differences in color in the current study.  Additionally, all l*a*b* values were had a weak positive correlation to backfat in the Page reference, so it’s unclear why the authros have fixated on b* from the current study, when L* and a* are much more relevant to meat quality as indicators of lightness and redness.  This whole discussion should be minimized or deleted all together.

Ln406-414: why were calculations done to report composition on a DM basis?  This is not a common practice, and it appears as though it was done to find a way to have differences in composition that were favorable to supplementation. 

Ln460-462: what is relevance on diet in the current study to the high proportion of oleic acid.  Oleic acid is normally the most abundant fatty acid, regardless of diet.

Ln520-527: There was only a tendency to improve growth and there were no differences in marbling, IMF, or shear force – the 3 main factors investigated on meat quality.  This summary must be revised to reflect the actual outcomes of the study and rectify the current inaccuracies. 

Round 2

Reviewer 1 Report

Authors did an important effort improving the manuscript. There are some comments that I previously did and you did not highlight them. I recommend you, to make the revision work easir, to hihglight in colours the changes for each reviewer. It helps for everyone. 

I do still thinking that the sample size is not strong enough

Moreover, I have some issues:

- I miss a sentence of introduction in the abstract. Manily something about BBC or why usinng or adding it to the feed...

- Albendazol is not an external anti-parasite drug. Please review and complete. Then, when you answer a reviewer comment, as you perfectly did, include the line or lines where you chaged it.

- Not using ultrasound scanner is an important lapse, please, say this limitation.

- Table 6, delete * and place the p value, as we see it is statistically different, we interpretate the superscripts.

- Now conlcusions are acording with the manuscript. 

Reviewer 2 Report

Ln24-26 & 625-626:  You still need to specify that IMF was higher only when adjusted to a DM basis – not the raw percentages.  That comment was not addressed sufficiently (at all) in the revised version. 

Moreover, your justification for adjusting to a DM basis was there were differences in moisture.  However, moisture percent was not different according to the values presented in Table 6.  Legitimate justification must be provided for reporting on DM basis because it seems as though it was done to promote numerical differences that were not statistically significant on a raw basis.  If no justification exists, then composition DM calculations should be removed and results/conclusions should be revised accordingly. 

Ln200: be consistent with abbreviation; the abbreviation is specified as LM on line 188. 

Ln200-203:  This sentence is very awkwardly worded and should be revised for English grammar.  Do you mean color was determined at 7 days postmortem?  How much time elapsed after ribbing before color data were collected?  In other words, how long was the muscle allowed to bloom/oxygenate?  Please provide the aperture size, observer angle, and light source for the colorimeter equipment. 
